# Optimization of Ultrasonic Extraction of Triterpenes from Loquat Peel and Pulp and Determination of Antioxidant Activity and Triterpenoid Components

**DOI:** 10.3390/foods11172563

**Published:** 2022-08-24

**Authors:** Yanwei Xue, Fei Wang, Chunhua Zhou

**Affiliations:** 1Department of Horticulture and Plant Protection, Yangzhou University, Yangzhou 225009, China; 2Joint International Research Laboratory of Agriculture & Agre-Product Safety of Ministry of Education of China, Yangzhou University, Yangzhou 225009, China

**Keywords:** loquat, ultrasonic extraction terpene, response surface, antioxidant, LC-MS

## Abstract

The aim of this paper was to study the optimal extraction process of total triterpenes from loquat peel and pulp assisted by ultrasound. The effects of solid–liquid ratio, ethanol concentration, ultrasonic time, ultrasonic power, and ultrasonic temperature on the yield of triterpenoid acid in loquat were investigated by single-factor and response surface methodology. FRAP (Ferric ion reducing antioxidant power) method, ABTS (2,2′-Azino-bis(3-ethylbenzthiazoline-6-sulfonic acid)) method, and DPPH (1,1-Diphenyl-2-picrylhydrazyl) method were used to determine the antioxidant capacity of peel and pulp at different stages. LC-MS (Liquid Chromatograph Mass Spectrometer) was used to qualitatively analyze different tissues of loquat. The optimal extraction conditions were as follows: ethanol concentration 71%, ultrasonic time 45 min, ultrasonic power 160 W, solid–liquid ratio 1:10, and ultrasonic temperature 30 °C. The total triterpenoid content of loquat peel was 13.92 ± 0.20 mg/g. The optimal extraction conditions were ethanol concentration 85%, ultrasonic time 51 min, ultrasonic power 160 W, solid–liquid ratio 1:8, and ultrasonic temperature 43 °C. The total triterpenoid content of loquat pulp was 11.69 ± 0.25 mg/g. The contents of triterpenes and antioxidant capacity in the peel and pulp of loquat at the three stages were the highest in the fruit ripening stage (S3). LC-MS analysis showed that most of the triterpenes belonged to ursolic acid derivatives and oleanolic acid derivatives, which laid the foundation for further utilization and development of loquat peel and pulp.

## 1. Introduction

Eriobotrya (*Eriobotrya japonica* Lindl.) is a common subtropical evergreen fruit tree found in South China. The fruit ripens in early summer, and the flesh is soft and juicy, sweet and sour, and has high nutritional value, which is favored by the market and consumers [1]. Loquat is a homologous plant whose flowers, fruits, leaves, roots, and white bark can all be used in medicine. Leaves of loquat are often used in traditional and modern medicine to treat coughs, etc. This fruit can quench thirst, inhibit vomiting and reverse circulation, and moisten the five zang organs [2,3].

Triterpenoid acids belong to the plant sterol family and are naturally bioactive components commonly found in cereals and vegetables [4]. In nature, loquat is a plant with high triterpenoid acid content [5]. Triterpenoid saponins in loquats mainly consist of ursolic acid, oleanolic acid, and corosolic acid, and belong to the pentacyclic family of triterpenoids which have high medicinal value [5,6] and are used as anti-inflammatory, anti-tumor, and antioxidation remedies, while also enhancing immunity [7,8]. Studies show that ursolic acid and oleanolic acid contents are high in loquat peel and have the potential to be processed as food or additives.

In recent years, research on the separation and extraction of triterpenoid acid compounds has become a hot topic at home and abroad. Currently, the separation methods of triterpenoid acids include Soxhlet extraction [9], cyclodextrin water extraction, ultrasonic extraction [10], the thermal reflux method (HRE) [11], and the Supercritical CO_2_ extraction method [12]. Compared to other methods, ultrasonic extraction, as an emerging extraction method, is widely favored by researchers due to its advantages of high extraction efficiency, low cost, and low energy consumption [13,14]. Its use of ultrasonic vibration can dissolve the required extract. Based on the acoustic principle, cavitation force as the main driving force can produce continuous compression under the action of a solvent. The formation of internal pressure microbubbles causes a “micro-explosion”. These produce small but significant shockwaves that produce subsequent releases of bioactive compounds from plant material.

Previous studies have shown that extraction efficiency can be affected by many factors, such as extraction temperature, time, and solid–liquid ratio [15], while the response surface method (RSM) can use multi-factor modeling to optimize statistical methods for complex processes. This method provides a free space in which experimental items can be defined according to response values, where the level of factors can be adjusted according to experimental requirements [16,17]. Therefore, in this study, the temperature, time, power, solid–liquid ratio, and ethanol concentration of ultrasonic extraction were optimized by a single-factor experiment. Based on this experiment, the response surface methodology was used to optimize the three most significant factors, and Box–Behnken experimental design with multiple quadratic regression was used to fit the functional relationship between factors and response values to determine the optimal parameters. Currently, there are many experiments and applications on loquat leaves, but there are few existing reports on the ultrasonic extraction of total triterpenoid acid from loquat peel and pulp. Therefore, it is of great significance to optimize the ultrasonic-assisted extraction of total triterpenoid acid from loquat peel and pulp. In this paper, using loquat peel and pulp as raw materials, combined with a single-factor test and response surface test, the optimal ultrasonic-assisted extraction conditions of loquat peel and pulp were evaluated, and the antioxidant capacity and component identification of loquat peel and pulp in different periods were studied for laying the foundation for efficient further development and utilization of loquat.

## 2. Materials and Methods

### 2.1. Chemicals and Instruments

-Anhydrous ethanol, glacial acetic acid, ethyl acetate, and other biochemical reagents and conventional reagents (analytically pure, Sinopharm Chemical Reagents Co., Shanghai, China).-Ursolic acid (analytically pure, concentration 98.5%, Shanghai Aladdin Biochemical Technology Co., Shanghai, China).-Vanillin (Shanghai Aladdin Biochemical Technology Co., Shanghai, China).-Total antioxidant capacity (FRAP-Trolox standard) kit (Suzhou Keming Biotechnology Co., Shanghai, China).-Total antioxidant capacity ABTS test box (Suzhou Keming Biotechnology Co., Shanghai, China).-1,1-Diphenyl-2-picrylhydrazyl (Suzhou Keming Biotechnology Co., Shanghai, China).-Mh-16kr Micro high speed refrigerated centrifuge (Shanghai Cheng Ke Instrument Co., Shanghai, China).-Tissuelyser-32 automatic sample rapid grinding instrument (Shanghai Jingxin Industrial Development Co., Shanghai, China).-Jp-040st Ultrasonic cleaning instrument (Shenzhen Jiemeng Cleaning Equipment Co., Shanghai, China).-Bhs-2 Electric Thermostatic Water Bath (Ningbo Qunan Experimental Instrument Co., Ningbo, China).-721 Visible Spectrophotometer (Shanghai Youke Instrument Co., Shanghai, China).-BSA124S electronic analysis balance (Beijing sartorius scientific instrument Co., Beijing, China).

### 2.2. Plant Materials

The “Bai Yu” loquat planted at the China Suzhou Evergreen Fruit Research Institute (Suzhou, China) was selected as plant material and collected on 6, 16, and 26 May 2021. These dates cover the transition from fruit swelling stage to fruit ripening stage, a total of 20 days, denoted as S1, S2, and S3. After the loquat was obtained, the peel and pulp were separated and wrapped in tin foil and stored at −80 °C for later use.

### 2.3. Methods

#### 2.3.1. Preparation of Test Solution

The preserved loquat peel and pulp were ground into a powder with a grinder, 0.2 g of each was weighed, and 3 mL of 75% ethanol was added. After uniform shaking, ultrasonic extraction was conducted at an ultrasonic temperature of 50 °C and an ultrasonic power of 240 W for 50 min. After cooling and centrifugation at 4000 rpm for 15 min, the supernatant was taken. The supernatant volume was increased to 30 mL with 75% ethanol for later use.

#### 2.3.2. Preparation of Ursolic Acid Standard

To prepare the ursolic acid, 20 mg standard ursolic acid was weighed, then dissolved in anhydrous ethanol. Anhydrous ethanol was then added to a constant volume of 100 mL, then shook well to obtain a 0.2 mg/mL standard of ursolic acid mother liquor, which was then stored at 4 °C for later use.

#### 2.3.3. Drawing of the Standard Curve

Amounts of 0.1, 0.2, 0.3, 0.4, 0.5, and 0.6 mL of uronic acid standard mother solution were added to the test tube and placed in a 90 °C water bath with hot dry solvent. Then, 5% vanillin-glacial acetic acid and 0.2 mL and 0.8 mL perchloric acid were added, shock mixed, and put in a 60 °C constant temperature water bath for 25 min after a cold bath for 10 min to room temperature. Ethyl acetate to 5 mL was added, with absolute ethanol instead of uronic acid as the standard blank group. The absorbance was measured at 546 nm. Using the mother liquor volume absorbance as ordinate standard drawing standard, the equation is y = 2.0161x − 0.0191, *R*^2^ = 0.9984.

#### 2.3.4. Determination of Total Triterpenes 

An amount of 0.2 mL of the mother liquid of the prepared test product was put into a 10 mL centrifuge tube and heated to 90 °C in a water bath. Then, 5% vanillin-glacial acetic acid (now supplied) and 0.2 mL and 0.8 mL perchloric acid in turn were added, mixed, and put in a 60 °C constant temperature water bath for 25 min before a cold bath for 10 min to room temperature. Ethyl acetate was added to 5 mL and the absorbance was measured at 546 nm.

Total triterpene acid content of each treatment was measured as per the following formula:$$W=\frac{c\times v\times n}{m}$$

Formula *W* represents the total triterpene acid content where mg/g; c is the mass concentration of the total triterpene of loquat peel or pulp, mg/mL; v is the extraction solution volume, mL; n is the dilution multiple; and m is the raw material mass, g.

#### 2.3.5. Single-Factor Experiment

The effects of ultrasonic power, ultrasonic time, ultrasonic temperature, solid–liquid ratio, and ethanol concentration on the contents of total triterpenoid acid in loquat fruit peel and flesh were investigated. All experiments were repeated three times. The extraction process is as follows.

(1)Screening of ultrasonic power

Five loquat pericarp (pulp) powders were weighed at 0.2 g per portion, with a solid–liquid ratio of 1:15 (g/mL), ethanol solution of 75%, ultrasonic temperature of 50 °C, ultrasonic time of 50 min, and ultrasonic power of 160 W, 180 W, 200 W, 220 W, and 240 W, respectively. Total triterpenoids were extracted to determine the appropriate ultrasonic power.

(2)Screening of ultrasound time

The total triterpenoids were extracted from 5 loquat pericarps (pulp) with 0.2 g of each at a solid–liquid ratio of 1:15 (g/mL), ethanol solution of 75%, ultrasonic power of 240 W, and an ultrasonic time of 50 min at ultrasonic times of 20, 30, 40, 50, and 60 min, respectively. Total triterpenoids were extracted to determine the appropriate ultrasonic time.

(3)Screening of ultrasonic temperature

The total triterpenoids were extracted from 5 loquat pericarps (pulp), 0.2 g per pericarp, with a solid–liquid ratio of 1:15 (g/mL), ethanol solution of 75%, ultrasonic power of 240 W, and ultrasonic time of 50 min at ultrasonic temperatures of 20 °C, 30 °C, 40 °C, 50 °C, and 60 °C, respectively, Total triterpenoids were extracted to determine the appropriate ultrasonic temperature.

(4)Screening of solid–liquid ratio

Five loquat pericarps (pulp) were weighed at 0.2 g per piece, 75% ethanol solution, ultrasonic power 240 W, ultrasonic time 50 min, ultrasonic temperature 50 °C, respectively, with the ratio of solid to liquid of 1:5, 1:10, 1:15, 1:20, 1. Under the condition of 25 (g/mL), total triterpenoids were extracted to determine the appropriate solid–liquid ratio.

(5)Screening of ethanol concentration

The total triterpenoids were extracted from 5 loquat pericarps (pulp) with 0.2 g of each, and the solid–liquid ratio was 1:15 (g/mL). The ultrasonic power was 240 W, the ultrasonic time was 50 min, and the ultrasonic temperature was 50 °C. The total triterpenoids were extracted under the ethanol concentrations of 55%, 65%, 75%, 85%, and 95%, respectively. The optimum ethanol concentration was determined.

### 2.4. Response Surface Methodology

According to the single-factor experiment results, three independent variables of the response surface optimization experiment were determined by using the Box–Behnken central combined experiment design principle of Design Expert 8.0.6 (Stat-ease, Minneapolis, America).

In the loquat peel experiment, ultrasonic power (A), ultrasonic time (B), and ethanol concentration (C) were selected as the influencing factors to design the response surface test. In the loquat pulp experiment, ultrasonic time (A), ultrasonic temperature (B), and solid–liquid ratio (C) were selected as the influencing factors for the design response surface experiment. In the peel, the ultrasonic power is 160–240 W, the ultrasonic time is 20–60 min, and the ethanol concentration is 55–95%.

The ultrasonic time in pulp was 20–60 min, the ultrasonic temperature was 20–60 °C, and the solid–liquid ratio was 1:5–1:25.

### 2.5. Determination of Antioxidant Capacity of Extracts from Different Parts of Loquat

According to the optimized conditions of response surface methodology, the total triterpenes of loquat peel and pulp from three periods were determined, and the antioxidant capacity of the extracts was determined by three methods.

#### 2.5.1. Determination of Total Antioxidant Capacity by FRAP (Ferric Ion Reducing Antioxidant Power) Method

An amount of 50 μL of loquat peel and pulp extract was taken, 950 μL of FRAP working solution was added, thoroughly mixed, and the reaction time was 20 min. The absorbance value was measured at 593 nm, denoted as A1. The above procedure was repeated with 50 μL FRAP extract instead of the extract, denoted A2. The total antioxidant capacity of the samples was calculated according to the formula:$$\mathrm{Total}\;\mathrm{antioxidant}\;\mathrm{capacity}\;\left(\mu \mathrm{mol}\;\mathrm{Trolox}/\mathrm{mL}\right)\;=\;\left(A1-A2-0.0134\right)÷2.4832$$

#### 2.5.2. Determination of Total Antioxidant Capacity by ABTS Method

An amount of 50 μL of loquat peel and pulp extract was taken, 950 μL of ABTS working solution was added, and the mixture was thoroughly mixed. The absorbance value was measured at 734 nm, denoted as A1. The procedure was repeated with 50 μL ABTS extract instead of the extract, denoted A2. ABTS free radical scavenging rate was calculated according to the formula:$$\mathrm{ABTS}\;\mathrm{free}\;\mathrm{radical}\;\mathrm{scavenging}\;\mathrm{rate}\;\left(\%\right)\;=\;\left(A2-A1\right)÷A2\times 100\%$$

#### 2.5.3. Determination of Total Antioxidant Capacity by DPPH Method

An amount of 7.886 mg DPPH free radical was weighed, the absolute ethanol was fixed in a 100 mL brown volumetric flask, and the solution was sonicated for 30 s. Then, the absolute ethanol was added to the calibration line, 0.2 mol/L DPPH absolute ethanol solution was prepared, and the solution was shaken and stored away from light. 

An amount of 1 mL of loquat peel and pulp extract was thoroughly mixed with the prepared DPPH absolute ethanol solution, and the absorbance was measured at 517 nm after being static for 1 h in the dark, denoted as A1. Similarly, 1 mL of the extract was mixed with 1 mL of absolute ethanol, and then the absorbance was measured at rest for 1 h under light, which was denoted as A2. Then, 1 mL of 30% ethanol solution was thoroughly mixed with the prepared DPPH absolute ethanol solution, and the absorbance was measured in the same way as above, which was denoted as A3. DPPH clearance rate was calculated according to the formula:$$\mathrm{DPPH}\;\mathrm{free}\;\mathrm{radical}\;\mathrm{scavenging}\;\mathrm{rate}\;\left(\%\right)\;=\;\left[1-\left(A1-A2\right)÷A3\right]\times 100\%$$

### 2.6. Identification of LC-MS Components in Different Parts of Loquat

#### 2.6.1. Sample Processing

The samples were placed in the lyophilizer and freeze-dried in vacuum, and then ground to powder with a grinding instrument (30 Hz, 1.5 min). An amount of 50 mg powder was weighed and dissolved in 1.2 mL 70% methanol extract, the vortexed once every 30 min, each lasting 30 s, for a total of 6 vortices. After centrifugation (at a speed of 12,000 rpm for 3 min), the supernatant was absorbed and the samples were filtered with a micropore membrane (0.22 μm pore size) and stored in injection bottles for PLC-MS/MS analysis.

#### 2.6.2. Liquid Phase Conditions

(1)Chromatographic column: Agilent SB-C18 1.8 µm, 2.1 mm × 100 mm;(2)Mobile phase: phase A was ultrapure water (0.1% formic acid was added), phase B was acetonitrile (0.1% formic acid was added);(3)Elution gradient: the proportion of B phase was 5% at 0.00 min, then linearly increased to 95% at 9.00 min, maintained at 95% for 1 min; at 10.00–11.10 min, the proportion of B phase decreased to 5%, and the proportion of B phase was balanced at 5% for 14.00 min;(4)Flow rate 0.35 mL/min; column temperature 40 °C; the injection volume was 4 μL.

#### 2.6.3. Mass Spectrum Conditions

Electrospray ionization (ESI) was performed at 550 °C; ion spray voltage (IS) 5500 V (positive ion mode)/−4500 V (negative ion mode); ion source gas I (GSI), gas II (GSII), and gas curtain gas (CUR) were set to 50, 60, and 25 psi, respectively, and the collision-induced ionization parameter was set to high. The instrument was tuned and mass calibrated using 10 and 100 μmol/L polypropylene glycol solutions in QQQ and LIT modes, respectively. QQQ scanning uses MRM mode with collision gas (nitrogen) set to medium. The DP and CE of each MRM ion pair are completed by further optimization of declustering potential (DP) and collision energy (CE). A specific set of MRM ion pairs was monitored in each period based on the metabolites eluted within each period.

## 3. Results

### 3.1. Single-Factor Experiment (Loquat Peel)

#### 3.1.1. Effects of Different Treatments on the Extraction Rate of Total Triterpenes from Loquat Peel

Figure 1 shows that the total triterpenoid acid content in loquat peel increased with the increase in ultrasonic time, and reached the maximum value at 40 min. As the ultrasonic time continued to increase, the total triterpenoid acid content exhibited a downward trend (Figure 2A). The total triterpenoid acid content did not change significantly with the change in ultrasonic temperature between the range of 20 °C and 40 °C but decreased when the temperature was too high. The total triterpenoid acid content reached its maximum at 30 °C (Figure 2B). The total triterpenoid content did not change significantly with the increase in ultrasonic power in the range of 160 W–180 W, and the total triterpenoid content reached its maximum at 180 W. However, when the ultrasonic power reached 180 W, the total triterpenoid extraction rate of loquat peel decreased significantly with the increase in ultrasonic power (Figure 2C). The total triterpenoid acid increased significantly from 1:5 to 1:10 (g/mL) in the ratio of solid to liquid. The content of total triterpenoid acid did not change significantly with the increase in extraction solvent (Figure 2D). The total triterpenoid acid content increased significantly with increasing ethanol concentration in the range of 55–65% and decreased with increasing ethanol concentration in the range of 65–95%, reaching its maximum value when the ethanol volume fraction was 65% (Figure 2E).

#### 3.1.2. Optimization Results of Single-Factor Test

Based on the single-factor test results, the optimal extraction conditions of triterpenoids from loquat peel were as follows: ultrasonic time 40 min, ultrasonic temperature 30 °C, ultrasonic power 180 W, solid–liquid ratio 1:10 (g/mL), and ethanol concentration 65%. The three most significant factors were ultrasonic time, ultrasonic power, and ethanol concentration.

#### 3.1.3. Optimization by Response Surface Methodology

According to the results of a single-factor experiment, three independent variables of a response surface optimization experiment were determined, which were ultrasonic power (A), ultrasonic time (B), and ethanol concentration (C). The total triterpenoid content represents the response value, and a total of 17 test sites were evaluated.

#### 3.1.4. ANOVA and Significance Test

Regression fitting of the Table 1 data was performed using Design-Expert 8.0.6 software to obtain a quadratic polynomial regression equation with the total triterpene content (Y) as the objective function: (1)$$Y=12.15-1.42A+0.72B-0.955C-0.1775\mathrm{AB}-0.3275\mathrm{AC}+0.1125\mathrm{BC}+0.3178A^{2}-1.61B^{2}-1.4C^{2}$$

Analysis of response surface variance results is shown in Table 2.

According to the ANOVA analysis in Table 2, the model *F* = 36.24, *p* < 0.0001, and the established model is extremely significant (*p* < 0.0001). As shown in Table 2, the mismatch term (*F* = 1.45, *p* = 0.3549) is not significant. The linear coefficient (A, B, C), quadratic coefficient (A^2^, B^2^, C^2^), and interaction coefficient (AB, AC) were all significant (*p* < 0.05). The *R*^2^ value was 0.9790, indicating that the model and experimental data fit reasonably, and the values of *AdjR*^2^ (0.9520) and Pre *R*^2^ (0.8094) indicate a strong correlation between experimental and predicted values, indicating that the model is significant and statistically reliable.

The effect of each factor on the total triterpene content was ultrasonic power (A) > ethanol concentration and (C) > ultrasound time (B).

#### 3.1.5. Analysis of the Interaction Factors of the Response Surface Methodology

To further clarify the interaction of each factor, multiple regression analyses were performed with 0.6 Design-Expert 8 (Table 2, with the results shown in Figure 3). The response surface methodology can intuitively reflect the influence of pairwise factors on the response value. In the 3D model, the greater slope of the model indicates that the factor has more influence on the response value; the gentler the model, the more the factor has less influence on the response value. In the contour map, the shape tends to be round, so the interaction of the factor is not significant. When the shape is oval, the interaction of the two factors is significant.

#### 3.1.6. Response Surface Optimization Results

According to the model analysis in Figure 3, it can be intuitively seen that the interaction of different factors affects the total triterpenoid content in the loquat peel. The optimal extraction conditions are as follows: ultrasonic power 160 W, ultrasonic time 45.43 min, ethanol concentration 70.73%, and the predicted value of total triterpenoid acid content was 14.08 mg/g.

### 3.2. Single-Factor Experiment (Loquat Pulp)

#### 3.2.1. Effects of Different Treatments on the Extraction Rate of Total Triterpenes from Loquat Pulp

As shown in Figure 2, the total triterpenoid acid content in loquat pulp increased significantly corresponding to the increase in ultrasonic time within the range of 20–50 min, and reached the maximum value at 50 min. With the increase in ultrasonic time, the total triterpenoid acid content showed a downward trend (Figure 2F). The total triterpenoid acid content increased significantly with the increase in ultrasonic temperature between the range of 20–40 °C. When the temperature increased to 40 °C, the total triterpenoid extraction reached the maximum value. As the temperature continued to increase, the total triterpenoid extraction rate decreased (Figure 2G). When the ultrasonic power was 160–240 W, the extraction rate of total triterpenoid acid decreased with the increase in ultrasonic power, and the change was not significant after 180 W. However, the change reached the maximum value when the ultrasonic power was 160 W (Figure 2H). The total triterpenoid acid content increased with the increase in solid/liquid ratio in the range of 1:5–1:10 (g/mL). When the ratio of solid to liquid was 1:10, the total triterpenoid acid content reached the maximum value, and the total triterpenoid acid content decreased significantly as the extraction solvent continued to increase (Figure 2I). The total triterpenoid acid content did not change significantly with the increase in ethanol concentration in the range of 55%–75%. With the increase in ethanol concentration, the total triterpenoid acid content increased. The total triterpenoid acid content reached its maximum value when the ethanol concentration reached 85% (Figure 2J).

#### 3.2.2. Optimization Results of Single-Factor Test

Based on the single-factor test results, the optimal extraction conditions of triterpenoids from loquat peel were as follows: ultrasonic time 40 min, ultrasonic temperature 30 °C, ultrasonic power 180 W, solid–liquid ratio 1:10 (g/mL), and ethanol concentration 65%. The three most significant factors were: ultrasonic time, ultrasonic power, and ethanol concentration.

#### 3.2.3. Optimization of Loquat Pulp by Response Surface Methodology

Based on the results of the univariate experiment, the three independent variables of the response surface optimization experiment were determined as follows: ultrasonic time (A), ultrasonic temperature (B), and material–solution ratio (C). The total triterpene content is the response value. Combined with the Box–Behnken method, the experiments were designed as shown in Table 3.

Design-expert 8.0.6 software was used for regression fitting of the data in Table 3, and a quadratic multinomic regression equation with total triterpenoid content (Y) as the objective function was obtained:Y = 11.78 + 0.3850A + 0.1825B + 0.4025C + 0.2475AB − 0.1825AC − 0.2525BC − 0.3867A^2^ −0.7667B^2^ − 2.14C^2^.(2)

The response variance results are shown in Table 4.

Table 3 data were fitted using Design-Expert 8.0.6 software to obtain a quadratic multinomial regression equation with total triterpene content (Y) as the objective function:(3)$$Y=11.78+0.3850A+0.1825B+0.4025C+0.2475\mathrm{AB}-0.1825\mathrm{AC}-0.2525\mathrm{BC}-0.3867A^{2}-0.7667B^{2}-2.14C^{2}$$

According to the ANOVA analysis in Table 4, the model was F = 48.58, *p* < 0.0001, and the established model was significant (*p* < 0.0001) and statistically significant. As shown in Table 4, the misfit term is not significant (F = 0.1040, *p* = 0.9534), which indicates that the model has sufficient predictive correlation to explain the correlation between the independent variable and the dependent variable. In the first term, A and C were significant (*p* < 0.05), and in the second term, A^2^, B^2^, and C^2^ were significant (*p* < 0.05). The *R*^2^ value of 0.9842 indicates the reasonable fit of the model and experimental data, and the values of *AdjR*^2^ (0.9640) and Pre *R*^2^ (0.9589) indicate a good correlation between experimental and predicted values, indicating that the model is significant and statistically reliable.

The effect of each factor on the total triterpene content is material to liquid ratio (C) > ultrasonic time (A) > ultrasonic temperature (B).

#### 3.2.4. Analysis of the Interaction Factors of the Response Surface

In order to clarify the interaction of each factor, the analysis was carried out in the same way as the peel method. The results are shown in Figure 4.

#### 3.2.5. Response Surface Optimization Results

According to the analysis in Figure 4, the interaction of different factors on the total triterpenoid content in loquat pulp can intuitively be seen. The optimal extraction conditions of triterpenoid acid from loquat pulp were as follows: ultrasonic time 50.93 min, ultrasonic temperature 43.94 °C, and solid–liquid ratio 1:8. The predicted value of ultrasonic extraction of total triterpenoid acid from loquat peel was 11.91 mg/g.

### 3.3. Determination of Antioxidant Capacity

The contents of total triterpene acids in loquat peel and pulp in S1, S2, and S3 (6, 16, 26 May) were determined under the optimized conditions of response surface test. The results showed that (Figure 5) the total triterpenoid acid content of loquat peel and pulp reached the maximum at the S3 maturity stage. The antioxidant capacity of loquat peel and pulp in three stages was analyzed by FRAP method, ABTS method, and DPPH method, and it was found that the maximum value was reached in the S3 maturity stage.

### 3.4. LC-MS Component Analysis

The software Analyst 1.6.3 was used to process the mass spectrometry data. The following figure shows the total ion current (TIC) diagram of the mixed QC samples (i.e., the time diagram of the sum of the intensities of all ions in the mass spectrometry at each time point). The abscissa is the retention time (Rt) of metabolite detection, and the ordinate is the ion flow intensity (CPS, count per second) of ion detection. The result is shown in Figure 6.

Qualitative and quantitative analysis of the metabolites of the sample was carried out by mass spectrometry. The characteristic ions of each substance were screened out by triple four-stage rods, and the signal intensity (CPS) of the characteristic ions was obtained in the detector. The mass spectrometry file of the sample was opened by MultiQuant software, and the chromatographic peak integration and correction were performed. The peak area (area) of each chromatographic peak represents the relative content of its counterpart, and the integral data of all chromatographic peak areas were finally derived and stored. The relative content of triterpene acid in loquat peel was higher than that in pulp.

A total of 29 triterpenoids were detected in loquat pulp and 49 triterpenoids were detected in the peel (Table 5), which were mainly ursolic acid derivatives and oleanolic acid derivatives.

## 4. Discussion

During extraction, there were many factors affecting efficiency. When extracting total triterpenoid acid content from loquat peel and pulp by ultrasound, we found that too long of an ultrasonic time would decrease total triterpenoid acid content. It is speculated that long ultrasonic extraction times may change the structure of triterpenoids, affect the stability of some triterpenoids, and make the number of triterpenoids decrease. It has also been reported that when the concentration of ethanol is too high, the triterpenes extracted will be mixed with more impurities. At elevated temperatures, some triterpenes may be oxidized and may contain hydroxyl groups [18,19,20]. Therefore, in the extraction of loquat pulp and peel, we chose a relatively mild time and temperature to achieve the maximum extraction rate. An appropriate increase in the size of the solid–liquid ratio can increase the yield, but excessive solid–liquid ratio may lead to waste of solvents and increase the cost [19,21]. When optimizing the ultrasonic power, we found that high ultrasonic power would reduce the yield, and we speculated that high ultrasonic power might destroy the triterpenoid compound structure and reduce its yield.

The optimal extraction conditions were obtained through a response surface optimization test. Combined with the actual operation test, the optimal ethanol concentration, ultrasonic time, and ultrasonic power were 71%, 45 min, and 160 W, respectively, in the loquat peel test. In the loquat pulp experiment, the optimal ultrasonic time, ultrasonic temperature, and solid–liquid ratio were 51 min, 44 °C, and 1:8 (g/mL), respectively. In addition, the maximum triterpenoid content of ultrasonic extraction of loquat peel and flesh was predicted to be 14.08 mg/g and 11.91 mg/g, respectively, and the actual yield of loquat peel and flesh was 13.92 mg/g and 11.69 mg/g, respectively, which were consistent with the predicted values.

## 5. Conclusions

The single-factor test determined three factors that had significant influence on the ultrasound-assisted extraction for the response surface test. In the response surface test of loquat peel, ethanol concentration, ultrasonic time, and ultrasonic power were selected to study the influence on the content of the total triterpenoid acid. Results show that within the scope of the experiment, the influence of various factors on the content of triterpenoid size were in the order of ultrasonic power > ethanol concentration (A) (B) (C) > ultrasonic time. Combined with the single-factor experiment and response surface ultrasonic-assisted extraction, optimum process conditions of total triterpenoid acid were as follows: 71% ethanol concentration, ultrasonic time is 45 min, ultrasonic power 160 W, and material liquid of 1:10. The ultrasonic temperature was 30 °C, and the actual measured value was 13.92 ± 0.20 mg/g by confirmatory test.

Response surface tests of loquat pulp, ultrasonic time, ultrasonic temperature, and solid–liquid ratio were selected to study their effects on the content of total triterpenoid acid. The results show that the influence of various factors on triterpenoid content was in the order of ultrasonic temperature (B) > ultrasonic time (A) > solid–liquid ratio (C). Combined with the single-factor test and response surface test, the optimal ultrasonic extraction conditions of loquat pulp were as follows: ethanol concentration 85%, ultrasonic time 51 min, ultrasonic temperature 44 °C, ultrasonic power 160 W, and solid–liquid ratio 1:8. Under these conditions, the actual measured value was 11.69 ± 0.25 mg/g through a verification test.

In the test of antioxidant capacity, this study first determined the total triterpene acid content in loquat peel and pulp in three periods and found that the total triterpene content in loquat peel and pulp in the S3 period reached the maximum. The antioxidant capacity was measured and analyzed by FRAP, ABTS, and DPPH methods, and it was found that the maximum value was reached in the S3 stage. A total of 29 triterpenoids were identified in loquat pulp and 49 triterpenoids were identified in loquat peel by LC-MS. Most of them are derivatives of ursolic acid or oleanolic acid.

The optimized processes of the single-factor test and response surface test could increase the content of the total triterpenoid acid in the peel and flesh of loquat fruit by ultrasonic extraction, which provided a certain basis for the extraction and comprehensive utilization of total triterpenoid acid in loquat fruit peel and flesh.

## Figures and Tables

**Figure 1 foods-11-02563-f001:**
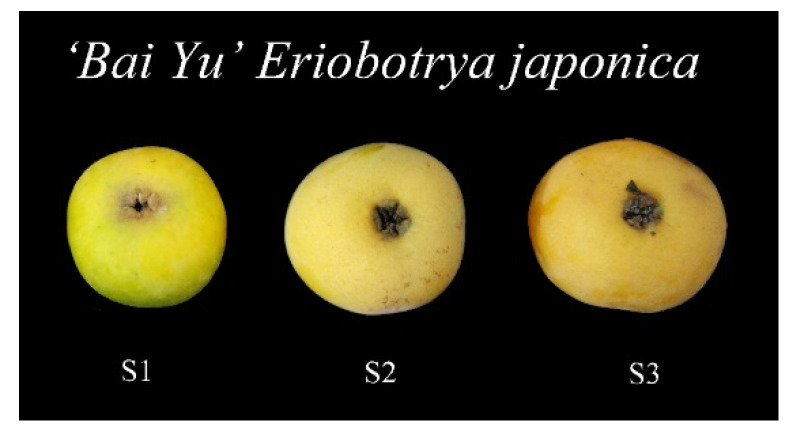
Loquat at different periods. Note: S1—6 May 2021, period S2—16 May 2021, and period S3—26 May 2021.

**Figure 2 foods-11-02563-f002:**
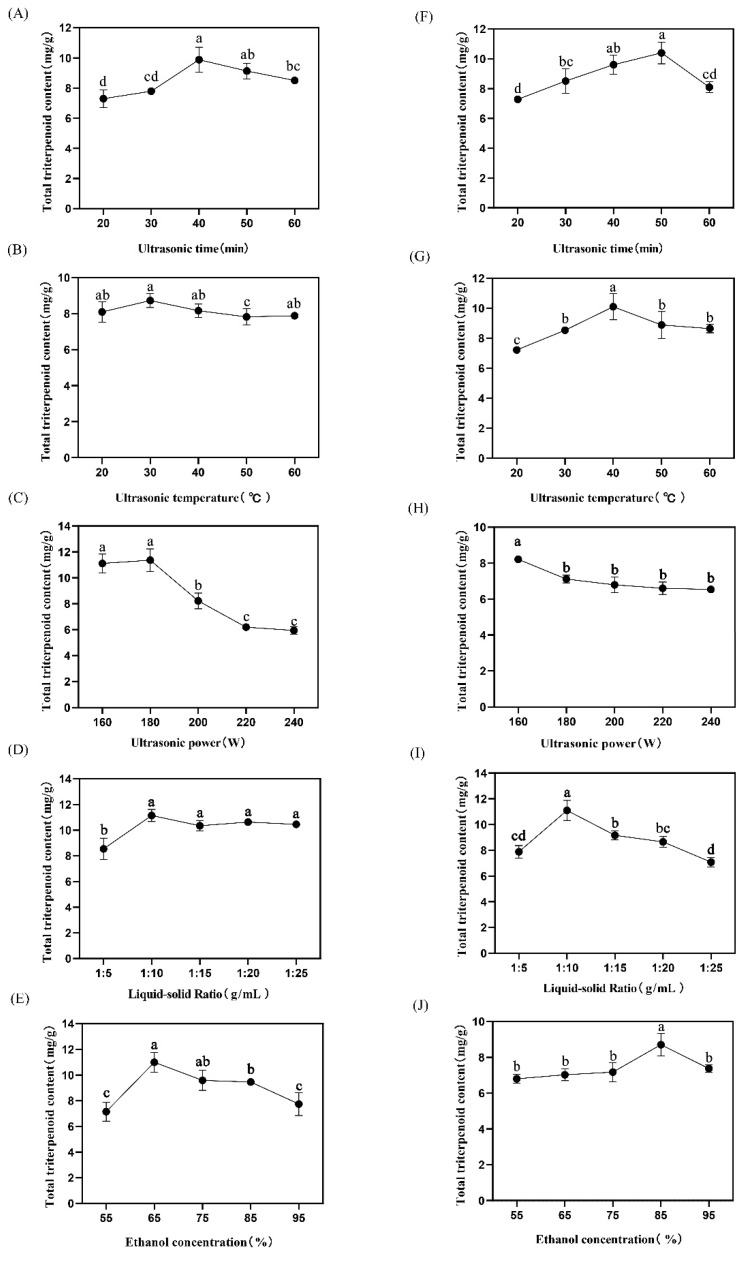
Effects of different factors on extraction of total triterpenes from loquat peel and pulp. (**A**–**E**): Effects of ultrasonic time, temperature, power, liquid-solid ratio and ethanol concentration on the yield of total triterpenes from loquat peel. (**F**–**J**): Effects of ultrasonic time, temperature, power, liquid-solid ratio and ethanol concentration on the yield of total triterpenes from loquat pulp. Different lowercase letters indicates significant difference (*p* < 0.05).

**Figure 3 foods-11-02563-f003:**
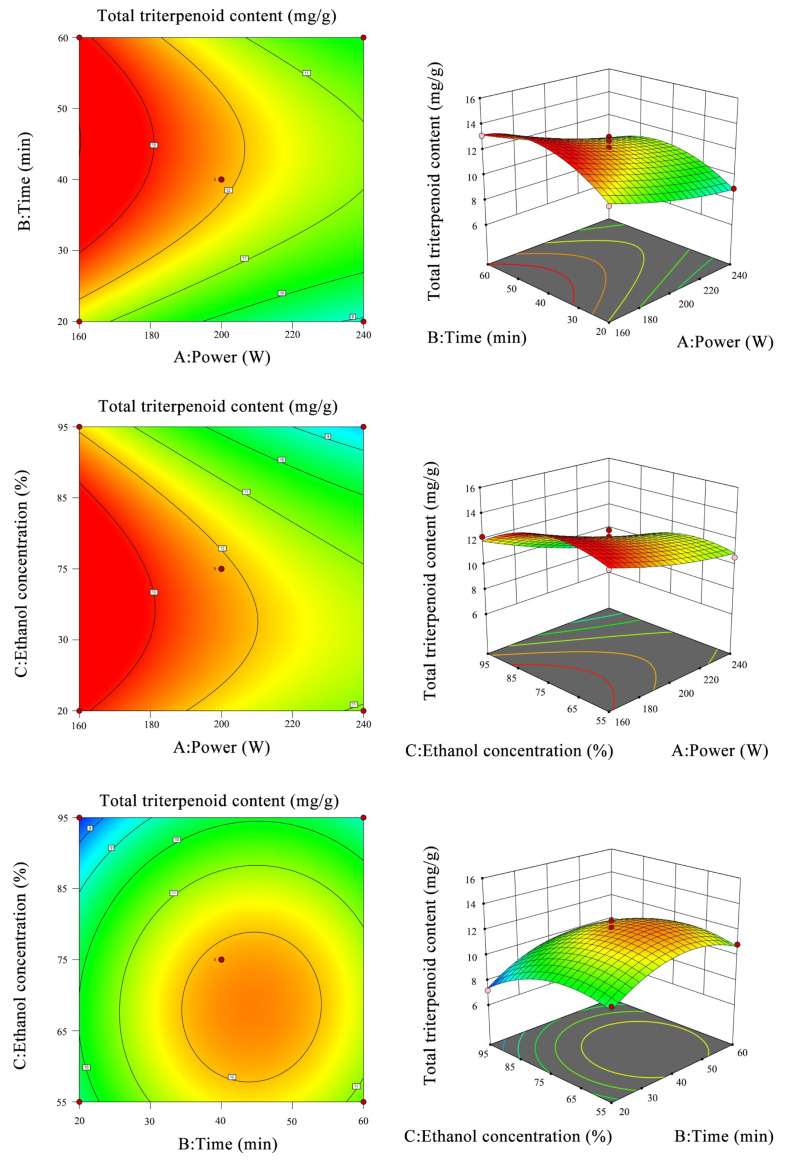
Response surface diagram and contour lines of different factors on total triterpenoid content in loquat peel.

**Figure 4 foods-11-02563-f004:**
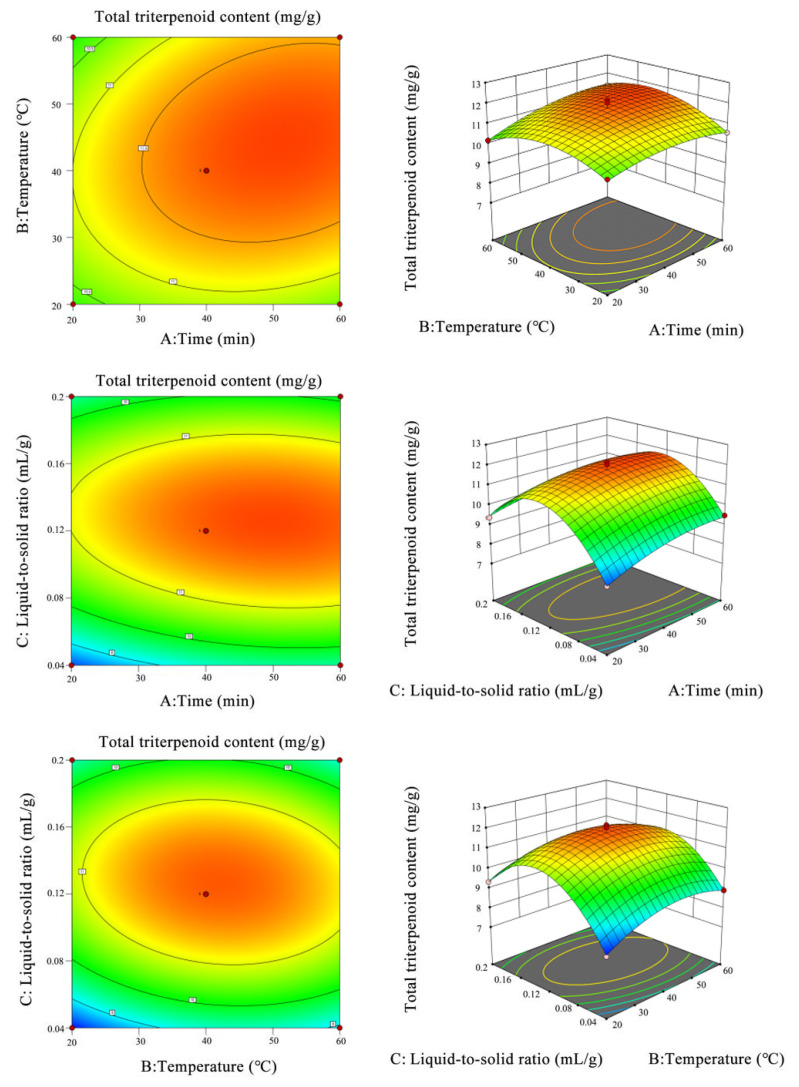
Response surface diagram and contour lines of different factors on total triterpenoid content in loquat pulp.

**Figure 5 foods-11-02563-f005:**
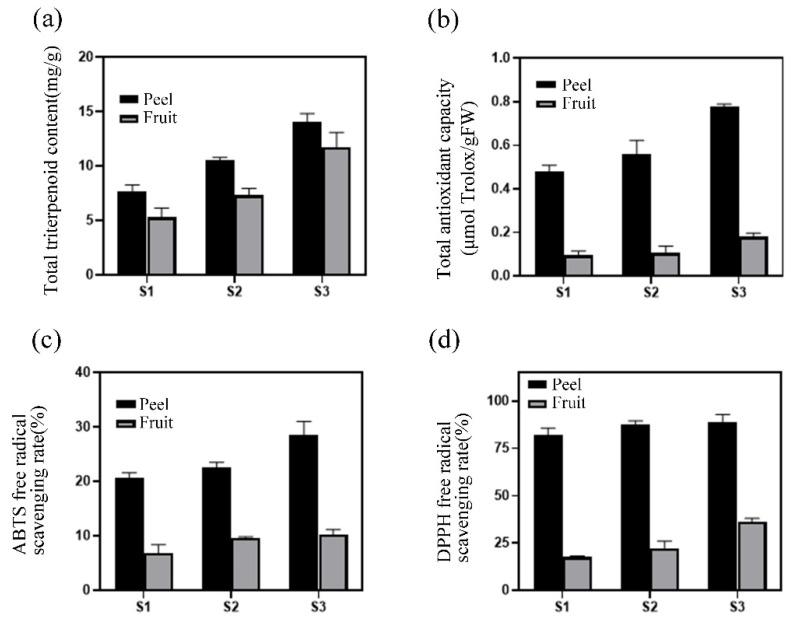
Analysis of antioxidant capacity: (**a**) total triterpenoid acid content; (**b**) total antioxidant capacity determination by FRAP method; (**c**) free radical scavenging rate determination by ABTS method; (**d**) determination of DPPH free radical scavenging rate.

**Figure 6 foods-11-02563-f006:**
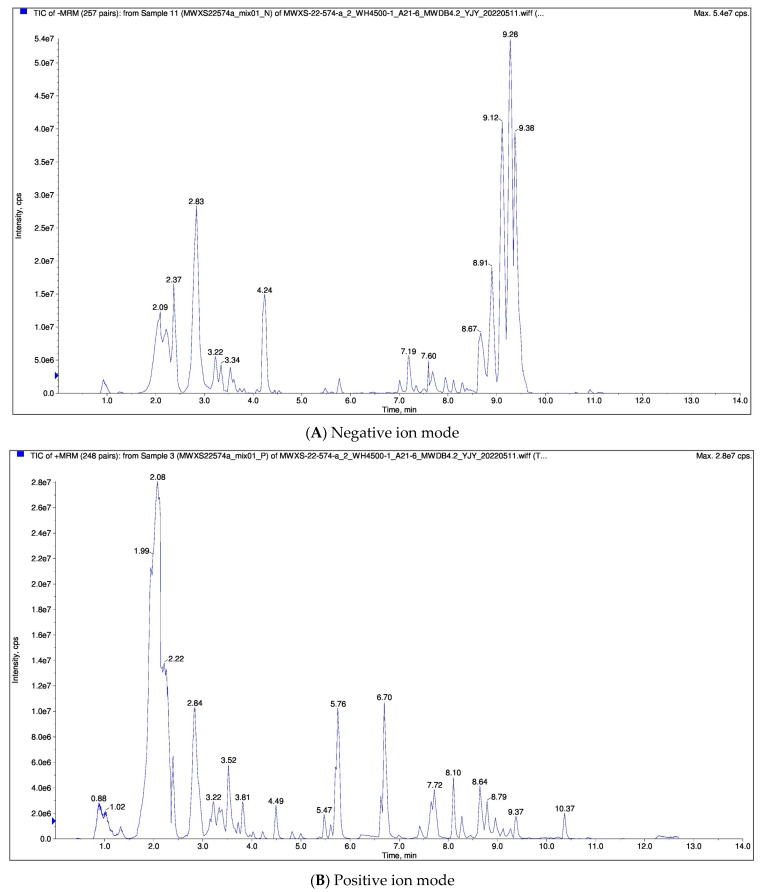
Total ion current diagram of mixing quality spectrum analysis.

**Table 1 foods-11-02563-t001:** Experimental results of response surface methodology (Loquat Peel).

Number	Factor			
	A: Ultrasonic Power (W)	B: Ultrasonic Time (min)	C: Ethanol Concentration (%)	Total Triterpenoid Content (mg/g)
1	240	40	55	10.59
2	200	40	75	12.22
3	200	60	95	8.74
4	200	40	75	11.97
5	200	20	55	9.78
6	240	60	75	10.15
7	200	20	95	7.19
8	240	40	95	8.48
9	200	40	75	11.87
10	160	40	95	12.21
11	200	40	75	12.74
12	240	20	75	8.95
13	200	60	55	10.88
14	160	40	55	13.01
15	160	60	75	13.13
16	200	40	75	11.96
17	160	20	75	11.22

**Table 2 foods-11-02563-t002:** Analysis of variance results (Loquat Peel).

Source of Variation	Sum of Square	Degree of Freedom	Mean Square	*F* Value	*p* Value
Model	48.54	9	5.39	36.24	<0.0001
A	16.25	1	16.25	109.17	<0.0001
B	4.15	1	4.15	27.87	0.0011
C	7.30	1	7.30	49.03	0.0002
AB	0.1260	1	0.1260	0.8469	0.3880
AC	0.4290	1	0.4290	2.88	0.1333
BC	0.0506	1	0.0506	0.3402	0.5780
A^2^	0.4251	1	0.4251	2.86	0.1348
B^2^	10.88	1	10.88	73.10	<0.0001
C^2^	8.22	1	8.22	55.24	0.0001
Residual	1.04	7	0.1488		
Lack of fit	0.5418	3	0.1806	1.45	0.3549
Pure error	0.4999	4	0.1250		
Total	49.58	16			

*R*^2^ = 0.9790; *AdjR*^2^ = 0.9520; Pred *R*^2^ = 0.8094. Notes: A—ultrasonic power; B—ultrasonic time; C—ethanol concentration.

**Table 3 foods-11-02563-t003:** Experimental results of response surface methodology (Loquat Pulp).

Number	Factor			
	A: Ultrasonic time (min)	B: Ultrasonic Temperature (°C)	C: Liquid–Solid Ratio (g/mL)	Total Triterpenoid Content (mg/g)
1	20	40	1:25	8.27
2	60	40	1:25	9.49
3	40	40	1:8.3	11.75
4	40	40	1:8.3	12.16
5	40	40	1:8.3	11.51
6	20	20	1:8.3	10.38
7	20	60	1:8.3	10.18
8	60	20	1:8.3	10.57
9	40	20	1:5	9.34
10	60	40	1:5	9.87
11	40	20	1:25	7.97
12	60	60	1:8.3	11.36
13	40	40	1:8.3	11.43
14	40	40	1:8.3	12.03
15	40	60	1:5	9.27
16	20	40	1:5	9.38
17	40	60	1:25	8.91

**Table 4 foods-11-02563-t004:** Analysis of variance results (Loquat Pulp).

Source of Variation	Sum of Square	Degree of Freedom	Mean Square	*F* Value	*p* Value
Model	27.14	9	3.02	48.58	<0.0001
A	1.19	1	1.19	19.10	0.0033
B	0.2664	1	0.2664	4.29	0.0770
C	1.30	1	1.30	20.88	0.0026
AB	0.2450	1	0.2450	3.95	0.0873
AC	0.1332	1	0.1332	2.15	0.1864
BC	0.2550	1	0.2550	4.11	0.0823
A^2^	0.6298	1	0.6298	10.14	0.0154
B^2^	2.48	1	2.48	39.87	0.0004
C^2^	19.22	1	19.22	309.66	<0.0001
Residual	0.4346	7	0.0621		
Lack of fit	0.0314	3	0.0105	0.1040	0.9534
error	0.4031	4	0.1008		
Total	27.58	16			

*R*^2^ = 0.9842; *AdjR*^2^ = 0.9640; Pred *R*^2^ = 0.9589. Notes: A—ultrasonic time; B—ultrasonic temperature; C—liquid–solid ratio.

**Table 5 foods-11-02563-t005:** Qualitative analysis by LC-MS.

Identification of Parts	Formula	Compounds	CAS
Peel, Fruit	C30H48O4	2α-Hydroxyursolic acid	-
Peel, Fruit	C30H46O5	2α,19α-Dihydroxy-3-oxours-12-en-28-oic acid	176983-21-4
Peel, Fruit	C30H48O4	Maslinic acid	4373-41-5
Peel, Fruit	C30H48O4	Alphitolic acid	19533-92-7
Peel, Fruit	C30H48O4	Corosolic acid	4547-24-4
Peel, Fruit	C30H48O4	3,24-Dihydroxy-17,21-semiacetal-12(13)oleanolic fruit	-
Peel, Fruit	C30H48O4	2,3-Dihydroxy-12-ursen-28-oic acid	-
Peel, Fruit	C30H48O5	Tormentic acid	13850-16-3
Peel, Fruit	C30H46O5	Rosamultic acid	214285-76-4
Peel, Fruit	C30H48O5	Euscaphic acid	53155-25-2
Peel, Fruit	C30H48O6	2α,3α,19α,23-tetrahydroxy-12-ursen-28-oic acid	-
Peel, Fruit	C30H48O4	Hederagenin	465-99-6
Peel, Fruit	C30H48O5	Arjunic acid	31298-06-3
Peel, Fruit	C30H48O5	Asiatic acid	464-92-6
Peel, Fruit	C30H46O4	Camaldulenic acid	71850-15-2
Peel, Fruit	C36H58O11	Nigaichigoside F1	95262-48-9
Peel, Fruit	C30H48O6	2α,3α,19α,23-Tetraydroxyurs-12-en-28-oic acid	-
Peel, Fruit	C30H48O6	2α,3α,19α-Trihydroxyursolic acid	-
Peel, Fruit	C30H48O6	1β,2α,3α,19α-Tetrahydroxyurs-12-en-28-oic acid	120211-98-5
Peel, Fruit	C30H48O6	Roxburic acid	108657-25-6
Peel, Fruit	C39H54O7	Caffeoylhawthorn acid	-
Peel, Fruit	C30H48O3	Oleanolic acid	508-02-1
Peel, Fruit	C30H48O6	1α,2β,3β,19α-Tetrahydroxyurs-12-en-28-oic acid	-
Peel, Fruit	C39H54O8	3-O-trans-cafeoyltormentic acid	-
Peel, Fruit	C39H54O7	3-O-cis-Coumaroyltormentic acid	-
Peel, Fruit	C40H56O8	3-O-Trans-feruloyl euscaphic acid	-
Peel, Fruit	C30H46O4	2α,3α-Dihydroxyurs-12,18-dien-28-oic acid	-
Peel, Fruit	C30H46O5	Swinhoeic acid	-
Peel, Fruit	C30H46O5	1-Oxo-Siaresinolic acid	-
Peel	C30H48O5	2α,3α,23-trihydroxyolean-12-en-28-oic acid	-
Peel	C39H54O7	3-O-trans-p-coumaroylrotundic acid	-
Peel	C30H44O5	Fupenzic acid	119725-20-1
Peel	C30H48O3	Ursolic acid	77-52-1
Peel	C30H48O3	(23S)-3β-hydroxydammara-21-oic acid 21,23-lactone	-
Peel	C30H46O6	2α,3α,19α-Trihydroxyurs-12-en-23-formyl-28-oic acid	-
Peel	C30H48O6	2α,3β,19α,23-Tetrahydroxyurs-12-en-28-oic acid	-
Peel	C30H48O6	1β,2α,3α-Trihydroxy-19-oxo-18,19-seco-urs-11,13(18)-dien-28-oic acid	-
Peel	C30H48O6	2α,3β,19α,23-Tetrahydroxyolean-12-en-28-oic acid	55306-03-1
Peel	C39H54O6	Jacoumaric acid	63303-42-4
Peel	C30H48O7	2α,3β,19α,23,24-Pentahydroxyolean-12-en-28-oic acid	-
Peel	C31H50O4	Corosolic Acid Methyl Ester	4518-70-1
Peel	C39H54O6	2α-hydroxy-3β-trans-p-hydroxycinnamoyloxy oleanolic acid	-
Peel	C39H54O6	3β-O-cis-p-Coumaroyl-2α-hydroxy-12-ursen-28-oic acid	-
Peel	C30H46O4	Pomonic acid	13849-90-6
Peel	C30H46O4	Hederagonic acid (23-Hydroxy-3-oxoolean-12-en-28-oic acid)	466-01-3
Peel	C30H44O5	3,11-Dioxo-19α-hydroxyurs-12-en-28-oic acid	-
Peel	C30H48O4	2α-hydroxyoleanolic acid	-
Peel	C39H54O6	p-Coumaroyleuscaphic acid	-
Peel	C36H58O10	Kajiichigoside F1(Euscaphic acid 28-O-β-D-glucopyranoside)	95298-47-8

## Data Availability

Data are contained within the article.

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
