# Peer review of "Optimization of Ultrasonic Extraction of Triterpenes from Loquat Peel and Pulp and Determination of Antioxidant Activity and Triterpenoid Components"

_foods, 2022, doi:10.3390/foods11172563_

Round 1

Reviewer 1 Report

In this manuscript, the authors present the optimization extraction process of triterpenoid compounds from loquat peel and pulp and determine the antioxidant capacity of peel and pulp by three methods (FRAP, ABTS, DPPH).

The following are the relevant requirements:

  1. Page 3: Please change the caption on the picture of the loquat in Figure 1 (5.6, 6.16, 5.26).
  2. Line 244: „3. Results and Discussion” should be „3. Results”

because in line 244 is „4. Discussion”.

  1. Please correct “References” according to “Instructions for Authors”.
  2. Lines 19 and 410: „Lc-ms” should be „LC-MS”.
  3. Line 46: „CO2” should be „CO2”.
  4. In the manuscript, there are no „Author Contributions” and „Conflict of Interest”.
  5. Between some words shall be filled with the space character (lines 22, 28, 98, 306, 361, 379, 383, 399).
  6. Between some words shall be removed with the space character (lines 162 and 167 should be "240W", line 231 should be 40°C).
  7. In the “Materials and Methods” section, the authors should include and explain the FRAP working solution, ABTS working solution and DPPH free radical.

Reviewer 2 Report

This study is about the optimization of ultrasound assisted extraction of triterpenes from s from loquat peel and pulp and determine the antioxidant activity.

The title of the MS should be revised. Terminology is wrong.

“determination of antioxidant activity” instead of “analysis of…”

And what type of components?

Please give a plausible explanation about how you have kept the temperature at 30oC during 45-50 min ultrasonic extraction. After 20 minutes the temperature rises dramatically.

Line 12: qualitatively? So, how did you determine the total content? Please indicate it.

Line 19: What do you mean by S3? Please write it in an open way.

Figure 2 and Figure 5 cannot be seen clearly. Please improve it.

However, I consider that this MS requires more work to be performed.

Round 2

Reviewer 2 Report

The authors have made the necessary changes and I suggest that it is ready for publication in its original form.